# Barriers in utilizing lipid-lowering agents in non-institutionalized population in the U.S.: Application of a theoretical framework

Abdullah A. Alfaifi[1]*, Leanne Lai[2], Abdullah U. Althemery[1]

**1** Department of Clinical Pharmacy, College of Pharmacy, Prince Sattam Bin Abdulaziz University, Al-Kharj, Kingdom of Saudi Arabia, **2** Department of Sociobehavioral and Administrative Pharmacy, College of Pharmacy, NOVA Southeastern University, Fort Lauderdale, Florida, United States of America

* a.alfaifi@psau.edu.sa

**Data Availability Statement:** The Medical Expenditure Panel Survey (MEPS) is a set of large-scale surveys of families and individuals, their medical providers, and employers across the

## Abstract

Cardiovascular diseases are a major cause of death globally. Epidemiological evidence has linked elevated levels of blood cholesterol with the risk of coronary heart disease. However, lipid-lowering agents, despite their importance for primary prevention, are significantly underused in the United States. The objective of this study was to explore associations among socioeconomic factors and the use of antihyperlipidemic agents in 2018 in U.S. patients with hyperlipidemia by applying a theoretical framework. Data from the 2018 Medical Expenditure Panel Survey were used to identify the population of non-institutionalized U.S. civilians diagnosed with hyperlipidemia. This cross sectional study applied the Andersen Behavioral Model to identify patients' predisposing, enabling, and need factors. Approximately 43 million non-institutionalized adults were diagnosed with hyperlipidemia. With the exception of gender and race, predisposing factors indicated significant differences between patients who used antihyperlipidemic agents and those who did not. The relation between income level and use of antihyperlipidemic agents was significant: $X^2$ (4, N = 3,781) = 7.09, $p$ <.001. Hispanic patients were found to be less likely to receive treatment (OR: 0.62; 95% CI: 0.43–0.88), as observed using a logistic model, with controls for predisposing, enabling, and need factors. Patients without health insurance were less likely to use lipid-lowering agents (OR: 0.33; 95% CI: 0.14–0.77). The present study offers essential data for prioritizing interventions by health policy makers by identifying barriers in utilizing hyperlipidemia therapy. Non-adherence to treatment may lead to severe consequences and increase the frequency of fatal cardiac events in the near future.

## Introduction

Cardiovascular diseases (CVD) are the major reason of death globally [1]. Based on World Health Organization (WHO) reports, stroke and coronary heart disease (CHD) account for the most global CVD deaths [2]. In the United States, the Centers for Disease Control and Prevention (CDC) has stated that heart disease is the leading cause of death. Heart diseases

United States. All data files available for downloading on the MEPS Web site are also available in the Agency for Healthcare Research and Quality Data Center. https://meps.ahrq.gov/mepsweb/.

**Funding:** This project was supported by the Deanship of Scientific Research at Prince Sattam Bin Abdulaziz University under the research project No. 2020/03/17208.

**Competing interests:** The authors have declared that no competing interests exist.

account for about 1 in every 4 deaths in the U.S, claiming 610,000 lives, with over half of these deaths due to CHD [3]. The expenses associated with heart diseases are enormous, exceeding 200 billion annually [3]. The total costs of CHD by itself amount to about $108.9 billion each year [3].

Epidemiological and clinical literature, including that obtained from the National Cholesterol Education Program (NCEP), has established the relationship between elevated levels of blood cholesterol and the risk of conditions, such as CHD and strokes [4, 5]. Patients with high blood cholesterol were more likely to develop heart disease [3]. It has been documented that for every reduction in cholesterol level, a reduction in the chance of CHD occurrence was observed [6]. The same pattern was observed in short-term clinical trials wherein for every 10% reduction in LDL, led to a similar percentage of reduced risk of CHD [4, 7].

National guidelines suggest lipid modification or the utilizing lipid-lowering agents for primary and secondary prevention of CHD and stroke. Guidelines on blood cholesterol management focused on β-Hydroxy β-methylglutaryl-CoA reductase inhibitor therapy as the primary choice for treatment [8, 9]. Besides statins, the addition of ezetimibe or a proprotein convertase subtilisin/kexin type (PCSK9) inhibitor was recommended for particular cases [9].

Despite the recommendations of all hyperlipidemia guidelines, only half of adults with hyperlipidemia received treatment [10]. This explains why only one-third of patients with high blood cholesterol had the condition under control [10]. To add fuel to the fire, about 29% and over 37% of adults with established CVD and diabetes, respectively, had not utilized any lipid-lowering medication [11].

According to the CDC, the utilization of lipid-lowering agents had varied due to different socioeconomic variables, for instance insurance coverage, race, age, and ethnicity [10, 12]. The results might signal that patients with high blood cholesterol faced barriers when trying to access lipid-lowering agents. Access to lipid-lowering agents is an important factor that explains the serious underutilization of such medications.

The Andersen behavior model was developed in the late 1960s to understand how and why families use health services. The model aimed to assist policy makers in promoting equitable access to healthcare. The model was previously utilized to explain access to different treatments [13–15]. The model suggests that patient's utilization of healthcare resources is determined by their predisposition to utilize services, factors that impede or enable their utilization, or their need for services. A limited number of studies have explored access to lipid-lowering medications for patients with high blood cholesterol in the U.S. without a theoretical framework.

The goal of this study was to explore the association between socioeconomic factors and the use of antihyperlipidemic agents in patients with hyperlipidemia in the U.S. in 2018 by applying the Andersen behavior model. This is an important goal for public health as the evidence shows how lipid lowering agent lowers CVDs [10, 11].

## Materials and methods

### Study data

The data are from the 2018 Medical Expenditure Panel Survey (MEPS). The MEPS is a nationally representative survey of the U.S. civilian non-institutionalized population, available since 1996. The MEPS data is publicly available and sponsored by the Agency for Healthcare Research and Quality.

## Study design

This is a population-based secondary data study, which used a cross-sectional design for the U.S. civilian non-institutionalized patients with hyperlipidemia to compare patients on therapy regimens comprising lipid-lowering medications to those on regimens without lipid-lowering medications in 2018. The model was developed after considering the literature. The model included population characteristics, including predisposing, enabling, and need factors, all factors that could impact patients' health behavior or their compliance for the use of lipid-lowering medications. The present study provided an unbiased overview of all groups of therapy used for hyperlipidemia.

## Study variables

Patients diagnosed with high blood cholesterol from the 2018 Medical Condition File formed the study population. These patients were obtained from the Agency for Healthcare Research and Quality's Clinical Classification Software Refined codes. The dependent variable (Hyperlipidemia therapy) was defined as a dichotomous variable: use of lipid-lowering agents versus no hyperlipidemia treatment, from the 2018 Prescribed Medicine File. The independent variables consisted of socioeconomic factors defined by the Andersen behavioral model of healthcare services [16, 17]. The variables include predisposing (gender, age, marital status, race, ethnicity, and education level), enabling (insurance coverage, income level, metropolitan area, and region), and need variables (self-perceived physical & mental health status, smoking, body mass index, hypertension, diabetes millets, stroke, and angina).

## Data analysis

A series of descriptive analyses using chi-square testing was conducted to assess differences in the sociodemographic factors of patients between lipid-lowering agents' users versus non-users. A multivariable logistic regression was calculated to assess the significant socioeconomic variables related with the utilization of treatment. Three logistic models were generated; the predisposing factors were compiled in the first model, the second model included enabling factors, and the third model added the need factors:

- Model 1: $\text{logitP}_1(X) = \alpha + \boldsymbol{\beta}_{predisposing} X_{predisposing}$

- Model 2: $\text{logitP}_2(X) = \alpha + \boldsymbol{\beta}_{predisposing} X_{predisposing} + \boldsymbol{\beta}_{enabling} X_{enabling}$

- Model 3: $\text{logitP}_3(X) = \alpha + \boldsymbol{\beta}_{predisposing} X_{predisposing} + \boldsymbol{\beta}_{enabling} X_{enabling} + \boldsymbol{\beta}_{need} X_{need}$

The study investigated adults diagnosed with high blood cholesterol from the U.S. adult civilian non-institutionalized population, a subgroup within the MEPS population. Estimates from the population of interest only yielded incorrect standard errors, usually overestimated standard errors, because of the multistage sample design MEPS follows [18]. The sample design was maintained by using the domain analysis, a subpopulation analysis. Domain analysis computes statistics for subgroups, but it accounts for the MEPS population when estimating variance for a subgroup. SAS 9.4 allows for the use of the subpopulation analyses required by a MEPS population.

## Results

Over 43 million noninstitutionalized adults (20 years of age and older) were diagnosed with hyperlipidemia in 2018. Among them, 86.90% received treatment (38,040,354) and 13.05% did not receive treatment (5,704,608). Table 1 presents the predisposing characteristics of patients

**Table 1. Population characteristics; predisposing factors.**

| Predisposing factors | Total number (unweighted) | Patients with hyperlipidemia treatment (unweighted) | Patients without hyperlipidemia treatment (unweighted) | p value |
|---|---|---|---|---|
| Age | | | | <.001* |
| 20–44 | 2,680,593 (190) | 1,853,030 (131) | 827,564 (59) | |
| 45–64 | 17,515,348 (1,423) | 15,113,962 (1,218) | 2,401,386 (205) | |
| 65 and older | 23,549,019 (2,168) | 21,073,362 (1,941) | 2,475,657 (227) | |
| Gender | | | | 0.260 |
| Male | 23,237,666 (1,891) | 20,352,319 (1,660) | 2,885,348 (231) | |
| Female | 20,507,294 (1,890) | 17,688,035 (1,630) | 2,819,260 (260) | |
| Race | | | | 0.190 |
| White | 35,630,231(2,976) | 31,176,376 (2,609) | 4,453,855 (367) | |
| Black | 4,420,464 (513) | 3,673,274 (422) | 747,191 (91) | |
| Others† | 3,694,265 (292) | 3,190,704 (259) | 503,561 (33) | |
| Ethnicity | | | | <.001* |
| Hispanic | 4,362,631 (480) | 3,408,099 (380) | 954,532 (100) | |
| Non-Hispanic | 39,382,330 (3,301) | 34,632,254 (2,910) | 4,750,075 (391) | |
| Marital status | | | | 0.004* |
| Married | 26,238,620 (2,137) | 23,137,335 (1,886) | 3,101,285 (251) | |
| Widowed | 5,892,207 (561) | 5,206,492 (494) | 685,715 (67) | |
| Others‡ | 11,614,134 (1,083) | 9,696,527 (910) | 1,917,607 (173) | |
| Education level | | | | 0.015* |
| Below High School | 4,856,305 (569) | 4,079,498 (480) | 776,808 (89) | |
| High School | 20,769,219 (1,844) | 17,857,413 (1,599) | 2,911,806 (245) | |
| Above High School | 18,119,436 (1,368) | 16,103,442 (1,211) | 2,015,993 (157) | |

* significance at 0.05 level

† others includes other race/ multiple race

‡ others includes single or separated

with hyperlipidemia. Fifty-three percent of the weighted study population was 65 years or older, and 40% was between 45–64. A little over half of subjects were male. The majority of patients were white and non-Hispanics. Sixty percent were currently married, and 41.42% had an education level above high school.

Table 2 describes the enabling factors about the current sample. Ninety-eight percent of the patients with hyperlipidemia were insured, with 62% of them holding a private insurance. Majority of the subjects reported a high-income level. Forty-four of hyperlipidemic patients receiving treatment have excellent and very good self-perceived physical health status compared to 35% of those without treatment (Table 3). The same phenomenon was observed with self-perceived mental health status, patients with treatment reported higher perceived mental status (59%) than patients without treatment (51%). Eighty-five percent patients with diabetes are on a antihyperlipidemic treatment regimen compared with 87.44% of non-diabetic patients.

Table 4 shows progressively adjusted logistic models of hyperlipidemic patients for receiving treatment. When only predisposing factors were included (model 1), individuals aged 20 to 44 and Hispanics were less prone to receive treatment than those aged 65 or more and non-Hispanics. After enabling factors were added (model 2), patients without insurance and patients living in the Midwest were less prone towards treatment compliance. On addition, the

**Table 2. Population characteristics; enabling factors.**

| Enabling Factors | Total number (unweighted) | Patients with hyperlipidemia treatment (unweighted) | Patients without hyperlipidemia treatment (unweighted) | p value |
|---|---|---|---|---|
| Insurance coverage | | | | <.001* |
| Any private | 27,079,592 (2,139) | 23,929,195 (1,892) | 3,150,397 (247) | |
| Public only | 16,081,633 (1,584) | 13,750,549 (1,358) | 2,331,084 (226) | |
| Uninsured | 583,735 (58) | 360,609 (40) | 223,126 (18) | |
| Income level | | | | <.001* |
| Poor/Negative | 4,399,802 (536) | 3,648,898 (448) | 750,904(88) | |
| Near poor | 1,552,884 (158) | 1,394,491 (139) | 158,394(19) | |
| Low income | 5,618,895 (536) | 4,661,304 (449) | 957,591 (87) | |
| Middle income | 11,578,220 (1,042) | 9,930,230 (904) | 1,647,990 (138) | |
| High income | 20,595,159 (1,509) | 18,405,430 (1,350) | 2,189,729 (159) | |
| Region | | | | 0.099 |
| Northeast | 7,992,648 (627) | 6,726,377 (535) | 1,266,270 (92) | |
| Midwest | 9,348,481 (827) | 8,319,684 (734) | 1,028,796 (93) | |
| South | 17,651,895 (1,535) | 15,343,415 (1,321) | 2,308,480 (214) | |
| West | 8,751,937 (792) | 7,650,877 (700) | 1,101,060 (92) | |

* significance at 0.05 level

Hispanic population between the ages of 20 to 44 was associated with likelihood of not receiving treatment.

The inclusion of need factors did not decrease the association between receiving treatment and patients aged 20 to 44, Hispanics, uninsured, and patients in the Midwest (model 3). However, none of the added need factors were significantly associated with antihyperlipidemic treatment.

## Discussion

This study aimed to provide an estimate of the prevalence of hyperlipidemia among noninstitutionalized adults in the U.S. civilian population in 2018, which is an estimated 13.4%. This figure is lower than that reported by the CDC, which estimated that 30% of the U.S. population had hyperlipidemia [19].

This study's estimation differed from the CDC's by including patients, wherein their hyperlipidemia report was later validated by a health care professional. In 2012, an estimated 72 million MEPS households self-reported a diagnosis of hyperlipidemia, indicating that the CDC's figures were drawn from self-reports rather than from reports validated by health care professionals [16]. Although the percentage was less than estimated, this study has already pointed out that more than 47 million U.S. adults have been diagnosed with hyperlipidemia, a main determinant reason for stroke and CHD.

This study has found that an alarming number of patients who had hyperlipidemia (5,704,608) did not use a lipid-lowering agent of any kind, reflecting the general national incidence of hyperlipidemia therapy in 2018. Indeed, this very shortfall instigated the present investigative study, for the Healthy People 2020 initiative aims to increase the prevalence of therapy that uses lipid-lowering agents alongside increasing adherence to such therapy [20].

Concerning the comparison of socioeconomic characteristics of patients who received treatment and patients who did not, this study obtained results similar to those obtained in previous studies. In 2012, for example, the percentage of U.S. adults who underwent

**Table 3. Population characteristics; need factors.**

| Need Factors | Total number (unweighted) | Patients with hyperlipidemia treatment (unweighted) | Patients without hyperlipidemia treatment (unweighted) | p value |
|---|---|---|---|---|
| Physical Health Status | | | | <.001* |
| Excellent | 5,459,491 (375) | 4,878,462 (325) | 581,030 (50) | |
| Very good | 14,258,527 (991( | 12,434,837 (848) | 1,823,691 (143) | |
| Good | 16,671,695 (1,331) | 14,440,573 (1,115) | 2,231,121 (216) | |
| Fair | 7,970,053 (827) | 6,362,455 (650) | 1,607,598 (177) | |
| Poor | 2,750,439 (273) | 2,105,671 (207) | 644,768 (66) | |
| Mental Health Status | | | | <.001* |
| Excellent | 12,261,144 (895( | 10,767,382 (758) | 1,493,763 (137) | |
| Very good | 14,154,464 (1039) | 12,524,093 (900) | 1,630,371 (139( | |
| Good | 15,325,710 (1,285) | 12,796,501 (1,035) | 2,529,209 (250) | |
| Fair | 4,410,820 (465) | 3,392,829 (363( | 1,017,991 (102) | |
| Poor | 958,067 (113) | 741,194 (89) | 216,873 (24( | |
| Diabetes | | | | |
| Yes | 14,425,574 (1,324) | 12,686,041 (1,118) | 1,739,534 (206) | |
| No | 32,684,631 (2,473) | 27,535,958 (2,027) | 5,148,673 (446) | |
| High Blood Pressure | | | | 0.394 |
| Yes | 33,938,952 (2,837) | 28,856,386 (2,337) | 5,082,566 (500) | |
| No | 13,171,253 (960) | 11,365,612 (808) | 1,805,641 (152( | |
| Angina | | | | 0.133 |
| Yes | 3,616,854 (281) | 3,201,838 (238( | 415,016 (43) | |
| No | 43,493,351 (3,516) | 37,020,160 (2,907) | 6,473,191 (609) | |
| Stroke | | | | 0.171 |
| Yes | 4,887,672 (448( | 4,018,182 (366( | 869,490 (82( | |
| No | 42,222,534 (3,349) | 36,203,816 (2,779) | 6,018,717 (570) | |
| Smoking Status | | | | 0.050 |
| Yes | 5,977,949 (508) | 4,894,193 (412) | 1,083,756 (96) | |
| No | 41,132,256 (3,289) | 35,327,805 (2,733) | 5,804,451 (556) | |
| Body Mass Index | | | | 0.028* |
| Underweight | 327,070 (28) | 233,341 (20) | 93,729 (8( | |
| Normal Weight | 10,194,145 (805) | 8,451,161 (657) | 1,742,984 (148) | |
| Overweight and Obese | 36,588,990 (2,964) | 31,537,496 (2,468) | 5,051,494 (496) | |

* significance at 0.05 level

hyperlipidemia therapy increased with age [11]. This study has identified a similar trend among patients who have hyperlipidemia, finding that the prevalence of diagnosis and treatment increases with age. This is an expected result considering that the atherosclerotic cardiovascular disease (ASCVD) risk estimator and the Framingham CVD risk calculator, used by clinicians to determine eligibility for hyperlipidemia therapy, treats age as a risk factor [21, 22]. A considerable number of hyperlipidemia patients aged 65 or older had not undergone treatment—2,475,657 altogether. Such a finding comports with the results of investigations made between 2005 and 2012, wherein a considerable number of patients aged 65 or older were found to have not opted for hyperlipidemia therapy despite being eligible for treatment [23].

Most young adults do not meet the criteria for hyperlipidemia therapy as set forth in national guidelines. Further, there is little consensus surrounding the use of lipid-lowering agents for the treatment of young patients. Thus some studies have concluded that early

**Table 4. Progressively adjusted logistic models of hyperlipidemic agents.**

| Odds Ratio Estimates | Model 1 | | | Model 2 | | | Model 3 | | |
|---|---|---|---|---|---|---|---|---|---|
| Effect | Point Estimate | 95% Wald Confidence Limits | | Point Estimate | 95% Wald Confidence Limits | | Point Estimate | 95% Wald Confidence Limits | |
| **Age** | | | | | | | | | |
| 20–44 | 0.28* | 0.18 | 0.44 | 0.27* | 0.17 | 0.42 | 0.27* | 0.170 | 0.44 |
| 45–64 age | 0.77 | 0.59 | 1.02 | 0.75 | 0.56 | 1.00 | 0.748 | 0.55 | 1.01 |
| 65 and older | Reference Group | | | Reference Group | | | Reference Group | | |
| **Gender** | | | | | | | | | |
| Female | 0.88 | 0.70 | 1.10 | 0.88 | 0.70 | 1.10 | 0.87 | 0.69 | 1.09 |
| Male | Reference Group | | | Reference Group | | | Reference Group | | |
| **Race** | | | | | | | | | |
| Black | 0.72 | 0.51 | 1.01 | 0.77 | 0.54 | 1.10 | 0.80 | 0.56 | 1.12 |
| Others | 0.94 | 0.56 | 1.57 | 1.02 | 0.63 | 1.65 | 1.01 | 0.63 | 1.64 |
| White | Reference Group | | | | | | Reference Group | | |
| **Ethnicity** | | | | | | | | | |
| Hispanic | 0.52* | 0.36 | 0.77 | 0.60* | 0.42 | 0.86 | 0.62* | 0.43 | 0.88 |
| Non-Hispanic | Reference Group | | | Reference Group | | | Reference Group | | |
| **Marital status** | | | | | | | | | |
| Widowed | 0.97 | 0.66 | 1.44 | 1.08 | 0.73 | 1.58 | 1.09 | 0.74 | 1.60 |
| Others | 0.82 | 0.63 | 1.07 | 0.91 | 0.68 | 1.21 | 0.90 | 0.68 | 1.20 |
| Married | Reference Group | | | Reference Group | | | Reference Group | | |
| **Education level** | | | | | | | | | |
| Below high school | 0.82 | 0.56 | 1.21 | 0.97 | 0.64 | 1.47 | 0.99 | 0.65 | 1.50 |
| High school | 0.80 | 0.62 | 1.03 | 0.86 | 0.66 | 1.12 | 0.88 | 0.67 | 1.15 |
| Above high school | Reference Group | | | Reference Group | | | Reference Group | | |
| **Insurance coverage** | | | | | | | | | |
| Public only | | | | 0.81 | 0.60 | 1.10 | 0.82 | 0.60 | 1.12 |
| Uninsured | | | | 0.33* | 0.14 | 0.77 | 0.33* | 0.14 | 0.77 |
| Any private | | | | Reference Group | | | Reference Group | | |
| **Income level** | | | | | | | | | |
| Low income | | | | 0.75 | 0.50 | 1.12 | 0.77 | 0.51 | 1.16 |
| Middle income | | | | 0.81 | 0.60 | 1.09 | 0.83 | 0.61 | 1.14 |
| Near poor | | | | 1.27 | 0.64 | 2.51 | 1.33 | 0.66 | 2.66 |
| Poor/negative | | | | 0.80 | 0.52 | 1.23 | 0.81 | 0.52 | 1.24 |
| High income | | | | Reference Group | | | Reference Group | | |
| **Region** | | | | | | | | | |
| Midwest | | | | 1.61* | 1.10 | 2.34 | 1.59* | 1.10 | 2.30 |
| South | | | | 1.35 | 0.97 | 1.87 | 1.33 | 0.96 | 1.85 |
| West | | | | 1.37 | 0.99 | 1.90 | 1.35 | 0.96 | 1.88 |
| Northeast | | | | Reference Group | | | Reference Group | | |
| **Physical health status** | | | | | | | | | |
| Fair | | | | | | | 0.87 | 0.50 | 1.50 |
| Good | | | | | | | 0.77 | 0.47 | 1.25 |
| Poor | | | | | | | 1.05 | 0.53 | 2.08 |
| Very good | | | | | | | 0.93 | 0.56 | 1.55 |
| Excellent | | | | | | | Reference Group | | |
| **Mental health status** | | | | | | | | | |
| Fair | | | | | | | 0.96 | 0.55 | 1.67 |

*(Continued)*

**Table 4.** (Continued)

| Odds Ratio Estimates | Model 1 | | | Model 2 | | | Model 3 | | |
|---|---|---|---|---|---|---|---|---|---|
| Effect | Point Estimate | 95% Wald Confidence Limits | | Point Estimate | 95% Wald Confidence Limits | | Point Estimate | 95% Wald Confidence Limits | |
| Good | | | | | | | 0.96 | 0.66 | 1.38 |
| Poor | | | | | | | 0.88 | 0.43 | 1.82 |
| Very good | | | | | | | 1.08 | 0.75 | 1.54 |
| Excellent | | | | | | | | | |
| **Diabetes** | | | | | | | | | |
| No | | | | | | | 1.01 | 0.78 | 1.31 |
| Yes | | | | | | | Reference Group | | |
| **Hypertension** | | | | | | | | | |
| No | | | | | | | 1.07 | 0.81 | 1.40 |
| Yes | | | | | | | Reference Group | | |
| **Angina** | | | | | | | | | |
| No | | | | | | | 1.01 | 0.70 | 1.45 |
| Yes | | | | | | | Reference Group | | |
| **Stroke** | | | | | | | | | |
| No | | | | | | | 1.03 | 0.72 | 1.48 |
| Yes | | | | | | | Reference Group | | |
| **Smoking** | | | | | | | | | |
| No | | | | | | | 0.90 | 0.66 | 1.23 |
| Yes | | | | | | | Reference Group | | |
| **Body Mass Index** | | | | | | | | | |
| Normal weight | | | | | | | 1.05 | 0.74 | 1.49 |
| Underweight | | | | | | | 1.36 | 0.71 | 2.64 |
| Overweight and obese | | | | | | | Reference Group | | |

* statistically significant

treatment of patients aged 35 to 55 prevents major CHD [24], however, others have recommended that younger adults focus on lifestyle modifications including smoking cessation and stepping up their level of physical activity [25]. Indeed, this study has found that most physicians take such an approach when treating younger adults.

Another important determinant of hyperlipidemia therapy was ethnicity, with Hispanics reporting less use of lipid-lowering agents than non-Hispanics. Some studies reported that Hispanics suffer relatively less mortality and morbidity resulting from CHD and CVD [26, 27], a trend that can potentially explaining underuse. However, reasoning could support the conclusion that Hispanics require less preventive treatment as they are somehow protected from heart disease [26].

One striking finding of comparative analysis has been the finding that comorbid conditions such as diabetes, hypertension, angina, and stroke are of no particular significance. Diabetes was expected to be a significant factor between other comorbid conditions. The 2013 ACC/AHA guidelines identified patients who have diabetes as a group whose members may benefit from statin treatment, a decision that may lead to explain the expectation. This is an alarm finding that patients, particularly with Hispanic origin, have barriers to treatment even with major comorbid conditions diagnoses.

Certainly the other perceived significance of comorbid conditions has varied in the literature. On one hand, McClelland and colleges found hypertension to significantly correlate with

the utilization of lipid-lowering medications in members of multiethnic groups who had atherosclerosis [28]. On the other hand, hypertension was found to significantly correlate with the use of or adherence to a regimen of lipid-lowering agents in adults and military veterans who had hyperlipidemia [29–31].

This study had some limitations; the cross-sectional design of MEPS—by its very nature—describes a limited period: the results of this study cannot be generalized to years other than 2018. Moreover, establishing causal relationships from a cross-sectional design presents a special set of difficulties. However, these limitations were not thought so great as to outweigh MEPS data's ability to supply various clinical and socioeconomic variables, generally hard to find in other databases. Although extensive studies linking socioeconomic variables and adherence to a specific lipid-lowering agent exist [32, 33], they failed to explore the relationship between socioeconomic factors and the use of all types of lipid-lowering medications, which included initiation of therapy.

## Conclusion

The objectives of the Healthy People initiative have changed over time, having focused initially on increasing cholesterol screening and subsequently on increasing treatments for patients with uncontrolled hyperlipidemia. This study serves as an evidence in the transition between these two focuses by highlighting factors significant for hyperlipidemia therapy. Moreover, this study draws its conclusions from real-world data obtained through controlled analysis. Beyond merely addressing the significance of a variety of socioeconomic factors for use of hyperlipidemia therapy, this study isolates certain particularly significant factors while controlling for other factors. In doing so, it provides data essential for prioritizing interventions by health policy makers.

## Acknowledgments

We would like to acknowledge and thank college of pharmacy at Prince Sattam bin Abdulaziz University for their administrative support.

## Author Contributions

**Conceptualization:** Abdullah U. Althemery.

**Data curation:** Abdullah U. Althemery.

**Formal analysis:** Leanne Lai.

**Funding acquisition:** Abdullah A. Alfaifi.

**Investigation:** Leanne Lai.

**Methodology:** Abdullah U. Althemery.

**Project administration:** Abdullah A. Alfaifi.

**Software:** Leanne Lai.

**Validation:** Leanne Lai.

**Visualization:** Leanne Lai.

**Writing – original draft:** Abdullah A. Alfaifi, Abdullah U. Althemery.

**Writing – review & editing:** Abdullah A. Alfaifi.

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
