## [Decision Letter · Decision Letter 0]

23 Apr 2021

PONE-D-20-39424

Barriers in Utilizing Lipid-Lowering Agents for U.S. Civilians Non-Institutionalized Population: A Theoretical Approach

PLOS ONE

Dear Dr. Abdullah A Alfaifi,

Thank you for submitting your manuscript to PLOS ONE. We are sorry for the delay with reviewing of your work. The pandemic situation increased work-loading on scientists and clinicians who work in the health science field  and slowed reviewing process. I hope you understand that peer-reviewing is a voluntary (un-paid) process and peer-reviewers are loaded with other priority work at this time. As editors, we did our best to speed-up the reviewing process. 

After careful consideration, we feel that your study has merit but does not fully meet PLOS ONE’s publication criteria as it currently stands. Therefore, we invite you to submit a revised version of the manuscript that addresses the points raised during the review process.

We look forward to receiving your revised manuscript.

Kind regards,

Olga A Sukocheva, PhD

Academic Editor

PLOS ONE

Journal Requirements:

In your Data Availability statement, you have not specified where the minimal data set underlying the results described in your manuscript can be found. PLOS defines a study's minimal data set as the underlying data used to reach the conclusions drawn in the manuscript and any additional data required to replicate the reported study findings in their entirety. All PLOS journals require that the minimal data set be made fully available. For more information about our data policy, please see http://journals.plos.org/plosone/s/data-availability.

Additional Editor Comments :

We appreciate your choice to submit your manuscript to Plos One journal.

After careful examination of the manuscript, we invite you to revise your manuscript. All reviewers' and Editor's comments should be addressed.

It is necessary to provide more details in the introduction and discussion section in order to demonstrate the novelty of your research and potential future application of your research outcome.

More relevant references should be included. Although the research design is appropriate the methods should be adequately described to indicate what is the main difference of your study from the previous similar analyses.

Reviewers' comments:

Reviewer's Responses to Questions

**Comments to the Author**

1. Is the manuscript technically sound, and do the data support the conclusions?

Reviewer #1: Partly

Reviewer #2: Yes

2. Has the statistical analysis been performed appropriately and rigorously? 

Reviewer #1: Yes

Reviewer #2: Yes

3. Have the authors made all data underlying the findings in their manuscript fully available?

Reviewer #1: Yes

Reviewer #2: Yes

4. Is the manuscript presented in an intelligible fashion and written in standard English?

Reviewer #1: Yes

Reviewer #2: Yes

5. Review Comments to the Author

Reviewer #1: This study describes a theoretical approach used to identify predisposing and enabling factors which impact use of lipid lowering drugs in USA population. Authors adapted the Andersen Behavioral Model. The study analyzed associations among socioeconomic factors and the use of antihyperlipidemic agents using data from the 2018 Medical Expenditure Panel Survey. The study aim is not novel and does not add any new knowledge. However, it does confirm previously found associations. There are many problems with this study.

1. Authors stated that (line 84)…”no study has explored access to lipid-lowering medications for patients with high blood cholesterol in the United States”. This is not entirely true. Those studies below addressed similar questions:

• Navar AM, Taylor B, Mulder H, Fievitz E, Monda KL, Fievitz A, Maya JF, López JAG, Peterson ED. Association of Prior Authorization and Out-of-pocket Costs With Patient Access to PCSK9 Inhibitor Therapy. JAMA Cardiol. 2017 Nov 1;2(11):1217-1225. doi: 10.1001/jamacardio.2017.3451. PMID: 28973087; PMCID: PMC5963012.

• Whayne TF. Outcomes, Access, and Cost Issues Involving PCSK9 Inhibitors to Lower LDL-Cholesterol. Drugs. 2018 Mar;78(3):287-291. doi: 10.1007/s40265-018-0867-9. PMID: 29396831.

• Salami JA, Warraich HJ, Valero-Elizondo J, Spatz ES, Desai NR, Rana JS, Virani SS, Blankstein R, Khera A, Blaha MJ, Blumenthal RS, Katzen BT, Lloyd-Jones D, Krumholz HM, Nasir K. National Trends in Nonstatin Use and Expenditures Among the US Adult Population From 2002 to 2013: Insights From Medical Expenditure Panel Survey. J Am Heart Assoc. 2018 Jan 22;7(2):e007132. doi: 10.1161/JAHA.117.007132. PMID: 29358195; PMCID: PMC5850149.

Notably, Salami et al., (2018) were using the same data base for their analysis. None of the above refs were cited.

2. Another problem with the study presentation. Authors claimed in the title that the study is a theoretical approach, while it is a cross-sectional data analysis ( indicated in Methods). So , the title is confusing. There is no information about the study type in the abstract. Cross -sectional type of study should be reflected clearly. The whole study should be adjusted to the cross-sectional type. Discussion should accent this.

3. It is indicated (line 95) that “The MEPS sample is a subsample of the National Health Interview Survey (NHIS) that the National Center for Health Statistics conducts.” – however, there is no link to the data. Is it publicly available data? Or access was granted? It is not indicated; no links provided.

4. Reference presentation is not appropriate/correct the reference list according to the journal requirements.

5. Study limitations are missing and should be described.

Reviewer #2: The manuscript by Alfaifi et al "Barriers in Utilizing Lipid-Lowering Agents for U.S. Civilians Non-Institutionalized Population: A Theoretical Approach." performed important theoretical study and relevant to the current scenario. The design and organization of the manuscript is well and noted worthy. authors used good English. I recommend the manuscript for publication in current form.

6. PLOS authors have the option to publish the peer review history of their article (what does this mean?). If published, this will include your full peer review and any attached files.

Reviewer #1: No

Reviewer #2: **Yes: **NAGENDRA YARLA

---

## [Author Response · Author response to Decision Letter 0]

27 Jun 2021

Dear Academic Editor: Olga A Sukocheva

Thank you for giving me the opportunity to submit a revised draft of my manuscript titled “Barriers in Utilizing Lipid-Lowering Agents for U.S. Civilians Non-Institutionalized Population: Application of Theoretical Framework” to PLOS ONE. 

I appreciate the time and effort that you and the reviewers dedicated to providing your valuable feedback on my manuscript. I am grateful to the reviewers for their insightful comments on this paper. I have been able to incorporate changes to reflect most of the suggestions provided by the reviewers. I have highlighted these changes in the manuscript. 

Here is a point-by-point response to the reviewers’ comments and concerns.

Comments from Reviewer 1

Comment 1: The study aim is not novel and does not add any new knowledge. However, it does confirm previously found associations. There are many problems with this study.

1. Authors stated that (line 84) …”no study has explored access to lipid-lowering medications for patients with high blood cholesterol in the United States”. This is not entirely true. Those studies below addressed similar questions:

• Navar AM, Taylor B, Mulder H, Fievitz E, Monda KL, Fievitz A, Maya JF, López JAG, Peterson ED. Association of Prior Authorization and Out-of-pocket Costs With Patient Access to PCSK9 Inhibitor Therapy. JAMA Cardiol. 2017 Nov 1;2(11):1217-1225. doi: 10.1001/jamacardio.2017.3451. PMID: 28973087; PMCID: PMC5963012.

• Whayne TF. Outcomes, Access, and Cost Issues Involving PCSK9 Inhibitors to Lower LDL-Cholesterol. Drugs. 2018 Mar;78(3):287-291. doi: 10.1007/s40265-018-0867-9. PMID: 29396831.

• Salami JA, Warraich HJ, Valero-Elizondo J, Spatz ES, Desai NR, Rana JS, Virani SS, Blankstein R, Khera A, Blaha MJ, Blumenthal RS, Katzen BT, Lloyd-Jones D, Krumholz HM, Nasir K. National Trends in Nonstatin Use and Expenditures Among the US Adult Population From 2002 to 2013: Insights From Medical Expenditure Panel Survey. J Am Heart Assoc. 2018 Jan 22;7(2):e007132. doi: 10.1161/JAHA.117.007132. PMID: 29358195; PMCID: PMC5850149.

Notably, Salami et al., (2018) were using the same data base for their analysis. None of the above refs were cited. 

Thank you for pointing this out. 

We agree that the novelty statement might be strong despite the fact that the examples are articles that looked at a particular antinyperlipidemic agent rather than the whole group. To address the issue, we modified the statement:

 In line 84: “Until now, no study has explored access to lipid-lowering medications for patients with high blood cholesterol in the United States” changed to “Until now, limited number of studies have explored access to lipid-lowering medications for patients with high blood cholesterol in the United States without a theoretical framework.”

Comment 2: Another problem with the study presentation. Authors claimed in the title that the study is a theoretical approach, while it is a cross-sectional data analysis (indicated in Methods). So, the title is confusing. There is no information about the study type in the abstract. Cross -sectional type of study should be reflected clearly. The whole study should be adjusted to the cross-sectional type. Discussion should accent this.

Thank you for bringing this up; it is true that the wording “theoretical approach” in the title might cause a confusion for the readers. To address this issue we have made two major modifications:

• We changed the title from “Barriers in Utilizing Lipid-Lowering Agents for U.S. Civilians Non-Institutionalized Population: Theoretical Approach” to “Barriers in Utilizing Lipid-Lowering Agents for U.S. Civilians Non-Institutionalized Population: Application of a Theoretical Framework”

• In the abstract (Line 29): We modified the following sentence: “A theoretical approach with which to identify patients’ predisposing, enabling, and need factors was adapted from the Andersen Behavioral Model” to “This is a cross sectional design study that applied the Andersen Behavioral Model to identify patients’ predisposing, enabling, and need factors.”

Comment 3: It is indicated (line 95) that “The MEPS sample is a subsample of the National Health Interview Survey (NHIS) that the National Center for Health Statistics conducts.” – however, there is no link to the data. Is it publicly available data? Or access was granted? It is not indicated; no links provided.

As we worked with MEPS for an extended period of time, we assumed all readers would have similar background. For clarification, MEPS is publicly available data, and the sample is a subsample of the National Health Interview Survey (NHIS). The link for the two datasets are publicly available.

In the manuscript, we changed the sentence to (line 98) “The MEPS data is publicly available and sponsored by The Agency for Healthcare Research and Quality.” 

Comment 4: Reference presentation is not appropriate/correct the reference list according to the journal requirements.

We apologize for the formatting errors (mentioned in comment 4). We have made the necessary changes to follow the reference style required by PLOS ONE. Moreover, we utilized a Scientific English editor to validated the manuscript style.

List of modifications for the references: 

• Used the recommended abbreviations for journals outlined in the ICMJE sample references. 

• Replaced all references that included bracket “[Internet]” with the actual links 

• Modified reference 9 to: Grundy SM, Stone NJ, Bailey AL, Beam C, Birtcher KK, Blumenthal RS, Braun LT, De Ferranti S, Faiella-Tommasino J, Forman DE, Goldberg R. 2018 AHA/ACC/AACVPR/AAPA/ABC/ACPM/ADA/AGS/APhA/ASPC/NLA/PCNA guideline on the management of blood cholesterol: a report of the American College of Cardiology/American Heart Association Task Force on Clinical Practice Guidelines. Journal of the American College of Cardiology. 2019 Jun 25;73(24):e285-350.

• Modified reference 12 to: Berger JH, Chen F, Faerber JA, O'Byrne ML, Brothers JA. Adherence with lipid screening guidelines in standard-and high-risk children and adolescents. American Heart Journal. 2021 Feb 1;232:39-46.

• Modified reference 20 to: Etats-Unis. Department of health and human services, Centers for Disease Control and Prevention, National Center for Health Statistics. Healthy people 2010: Final review. US Government Printing Office; 2012.

Comment 5: Study limitations are missing and should be described.

Addressed, the following paragraph was added at the end of the discussion session 

“This study had some limitations; the cross-sectional design of MEPS—by its very nature—describes a limited period: the results of this study cannot be generalized to years other than 2018. Moreover, establishing causal relationships from a cross-sectional design presents a special set of difficulties. However, these limitations were not thought so great as to outweigh MEPS data’s ability to supply various clinical and socioeconomic variables, generally hard to find in other databases. Although extensive studies linking socioeconomic variables and adherence to a specific lipid-lowering agent exist [32–33], they failed to explore the relationship between socioeconomic factors and the use of all types of lipid-lowering medications, which included initiation of therapy.”

Comments from Reviewer 2

Comment 1: The manuscript by Alfaifi et al "Barriers in Utilizing Lipid-Lowering Agents for U.S. Civilians Non-Institutionalized Population: A Theoretical Approach." performed important theoretical study and relevant to the current scenario. The design and organization of the manuscript is well and noted worthy. authors used good English. I recommend the manuscript for publication in current form.

Thank you for endorsing the manuscript. To ensure that the language is better, we have submitted the manuscript to a Scientific Manuscript Editing Service for academic papers. 

Comments from editorial 

It is necessary to provide more details in 1- the introduction and discussion section in order to demonstrate the novelty of your research and potential future application of your research outcome. 2- More relevant references should be included. 3- Although the research design is appropriate the methods should be adequately described to indicate what is the main difference of your study from the previous similar analyses.

1. We would like to thank the editors and the reviewers for their comments. In response to the first point raised by the editor, we added more details to the discussion to ensure the novelty of the study (Line 240 -249) 

2. As an extension to the first point, we added relevant references–suggested by reviewer 1 in the discussion (References number 32 & 33). 

3. An additional statement was added under the Methods study design (line: 112-114): “The present study provided an unbiased overview of all groups of therapy used for hyperlipidemia.”

Additional points: 

• The guidelines recommend that abbreviations may be used only if the term is mentioned more than 3 times. Line 69 was changed to “β-Hydroxy β-methylglutaryl-CoA” instead of “HMG CoA reductase inhibitor” 

• We used “The U.S.” consistently throughout the manuscript 

I look forward to hearing from you regarding my revised submission and in of further questions and comments you may have. 

Sincerely,

Abdullah Alfaifi 

Assistant Professor

Department of Clinical Pharmacy

Pharmacoeconomics and Outcome Research

Tel: +966 11-588- 6058

E-mail: a.alfaifi@psau.edu.sa

---

## [Decision Letter · Decision Letter 1]

23 Jul 2021

Barriers in utilizing lipid-lowering agents in non-institutionalized population in the U.S.: Application of a theoretical Framework

PONE-D-20-39424R1

Dear Dr. Alfaifi,

We’re pleased to inform you that your manuscript has been judged scientifically suitable for publication and will be formally accepted for publication once it meets all outstanding technical requirements.

Kind regards,

Olga A Sukocheva, PhD

Academic Editor

PLOS ONE

Additional Editor Comments (optional):

Reviewers' comments:

Reviewer's Responses to Questions

**Comments to the Author**

1. If the authors have adequately addressed your comments raised in a previous round of review and you feel that this manuscript is now acceptable for publication, you may indicate that here to bypass the “Comments to the Author” section, enter your conflict of interest statement in the “Confidential to Editor” section, and submit your "Accept" recommendation.

Reviewer #1: All comments have been addressed

Reviewer #2: All comments have been addressed

2. Is the manuscript technically sound, and do the data support the conclusions?

Reviewer #1: Yes

Reviewer #2: Yes

3. Has the statistical analysis been performed appropriately and rigorously? 

Reviewer #1: Yes

Reviewer #2: Yes

4. Have the authors made all data underlying the findings in their manuscript fully available?

Reviewer #1: Yes

Reviewer #2: Yes

5. Is the manuscript presented in an intelligible fashion and written in standard English?

Reviewer #1: Yes

Reviewer #2: Yes

6. Review Comments to the Author

Reviewer #1: I am satisfied with the revised version of this manuscript; authors addressed all my suggestions properly.

Reviewer #2: The manuscript by Alfaifi et al "Barriers in Utilizing Lipid-Lowering Agents for U.S. Civilians Non-Institutionalized Population: A Theoretical Approach." performed important theoretical study and relevant to the current scenario. The design and organization of the manuscript is well and noted worthy. authors used good English. I recommend the manuscript for publication in current form. Authors addressed questions and there is no concerns from my side.

7. PLOS authors have the option to publish the peer review history of their article (what does this mean?). If published, this will include your full peer review and any attached files.

Reviewer #1: No

Reviewer #2: **Yes: **NAGENDRA YARLA

---

## [Editor Report · Acceptance letter]

28 Jul 2021

PONE-D-20-39424R1 

Barriers in utilizing lipid-lowering agents in non-institutionalized population in the U.S.: Application of a theoretical Framework 

Dear Dr. Alfaifi:

I'm pleased to inform you that your manuscript has been deemed suitable for publication in PLOS ONE. Congratulations! Your manuscript is now with our production department. 

Kind regards, 

on behalf of

Dr. Olga A Sukocheva 

Academic Editor

PLOS ONE